# Involving Parents in Promoting Healthy Energy Balance-Related Behaviors in Preschoolers: A Mixed Methods Impact and Process Evaluation of SuperFIT

**DOI:** 10.3390/nu13051605

**Published:** 2021-05-11

**Authors:** Lisa S. E. Harms, Sanne M. P. L. Gerards, Stef P. J. Kremers, Kathelijne M. H. H. Bessems, Carsten van Luijk, Tülay Arslan, Femke M. Mombers, Jessica S. Gubbels

**Affiliations:** Department of Health Promotion, School of Nutrition and Translational Research in Metabolism (NUTRIM), Maastricht University, P.O. Box 616, 6200 MD Maastricht, The Netherlands; sanne.gerards@maastrichtuniversity.nl (S.M.P.L.G.); s.kremers@maastrichtuniversity.nl (S.P.J.K.); k.bessems@maastrichtuniversity.nl (K.M.H.H.B.); carstenvanluijk@gmail.com (C.v.L.); info@arslanzorg.nl (T.A.); f.mombers@student.maastrichtuniversity.nl (F.M.M.); jessica.gubbels@maastrichtuniversity.nl (J.S.G.)

**Keywords:** children, home, intervention, impact, nutrition, parents, physical activity, preschoolers, process evaluation

## Abstract

Parental involvement is an essential component of obesity prevention interventions for children. The present study provides a process and impact evaluation of the family component of SuperFIT. SuperFIT is a comprehensive, integrated intervention approach aiming to improve energy balance-related behaviors (EBRBs) of young children (2–4 years). A mixed methods design combined in-depth interviews with parents (*n* = 15) and implementers (*n* = 3) with questionnaire data on nutritional and physical activity-related parenting practices (CFPQ and PPAPP), the physical home environment (EPAO_SR) (*n* = 41), and intervention appreciation (*n* = 19). Results were structured using the concepts of reach, adoption, implementation, and perceived impact. Findings indicated that the families reached were mostly those that were already interested in the topic. Participants of the intervention appreciated the information received and the on-the-spot guidance on their child’s behavior. Having fun was considered a success factor within the intervention. Parents expressed the additional need for peer-to-peer discussion. SuperFIT increased awareness and understanding of parents’ own behavior. Parents made no changes in daily life routines or the physical home environment. Translating knowledge and learned strategies into behavior at home has yet to be achieved. To optimize impact, intervention developers should find the right balance between accessibility, content, and intensity of interventions for parents.

## 1. Introduction

Children’s dietary behavior, physical activity (PA) levels, and sedentary behavior, commonly called energy balance-related behaviors (EBRBs), contribute to the development of obesity [1]. Development of EBRB habits starts in early childhood and continues into adulthood [2,3,4]. The early promotion of healthy EBRBs is therefore key to preventing childhood overweight [5,6].

The home setting is a crucial setting to target preventive efforts, as most young children spend a lot of their time at home. Since parents are considered gatekeepers of the home setting [7,8], they can create a supportive environment, thereby promoting the development of healthy EBRBs. A supportive environment, for example, includes the availability and accessibility of water, fruit, vegetables, and play equipment [9,10,11,12] as well as positive parenting in the form of role modeling, parental support, and the use of an authoritative parenting style [13,14,15]. Finally, the establishment of rules and norms such as having joint family meals is also important in encouraging healthy EBRBs [16,17].

A well-designed intervention can help parents to create changes in the home environment [18]. However, reaching parents is challenging [19]. Schools and childcare organizations reach many children through their existing infrastructure [20], and thus are potential pathways for reaching both children and their parents. In addition, school or childcare EBRB interventions with parental involvement have been increasingly investigated [18,21,22,23,24], as settings do not operate in isolation but interact with other settings and intermediates [25]. In other words, the influence a parent has on a child’s behavior is affected by the way childcare staff handle similar situations, and vice versa [26,27]. Previous interventions focusing on childcare and parents have showed that actively involving parents increases the likelihood of success in influencing EBRBs in children [23].

SuperFIT is an intervention approach that was developed in the Netherlands and aims to promote healthy EBRBs in preschoolers [28]. SuperFIT targets both home and childcare through parents and childcare staff. Van de Kolk et al. (2020) provide a detailed description of the SuperFIT intervention approach. In brief, intervention strategies stimulate the formation of supportive environments through group sessions with parents, and the training and coaching of childcare staff. Evaluations of SuperFIT assessed effectiveness of parental involvement on child outcomes such as BMI-z score, physical activity, and dietary intake [29,30]. Effects on child outcomes are limited and seem to manifest on days that children attended childcare [29]. Effects on child outcomes on days at home seem to be lacking, despite extensive efforts to reach and involve parents [29,30].

Exploring the implementation process of the family component of SuperFIT can contribute to understanding why the intervention did not have the intended effects at home, and how the intervention can be optimized for parents [31]. Process evaluations can identify causal mechanisms or contextual factors related to outcome [32]. A process evaluation for SuperFIT’s childcare component has already been reported elsewhere [33]. Process evaluation studies of previous interventions targeting parents have shown the difficulty of engaging high-risk families [34] and the need to tailor the intervention to the abilities of parents [35]. The present study will focus on the processes in the home setting of participating families. Our attention will be directed toward perspectives of the implementers and parents to gain insight into what works for whom and why. With regard to the underlying processes, SuperFIT may have affected intermediate outcomes such as parental behavior or the home environment, neither of which formed part of the previous effect evaluations.

The present study reports the findings of a process and impact evaluation of the SuperFIT activities targeting the home setting. We based our findings on the Reach, Effectiveness, Adoption, Implementation, and Maintenance (RE-AIM) framework [36,37]. The following questions will be answered: (i) did we *reach* the intended target population and why (or why not)?; (ii) how was the family component delivered and *adopted*?; (iii) how did parents and implementers *experience the implementation* process of the family component?; and (iv) what is the perceived *impact* of participating in the family component on parental behavior, parenting practices, physical home environment, and the child EBRBs?

## 2. Materials and Methods

### 2.1. Design

The present study applied a process and impact evaluation of SuperFIT using a mixed methods design, combining quantitative and qualitative research among parents and intervention implementers. The study focused primarily on relevant processes during intervention activities for families and in the home setting. Effects on child outcomes have been previously assessed [29,30] and will not be reported in the present study.

### 2.2. SuperFIT Intervention Approach

SuperFIT is a comprehensive, integrated intervention approach aiming to improve the EBRBs of young children (2–4 years). SuperFIT is based on systems theory [26], targeting the most important micro-settings of a child—childcare and home—and aligning them with each other. In addition, it targets the different types of environments according to the ANGELO framework [38]. In the case of SuperFIT, home and childcare represent micro settings of the child in which physical (e.g., availability of play materials), sociocultural (e.g., positive parenting practices), political (e.g., family norms), and economic aspects (e.g., costs) influence the children’s EBRBs. SuperFIT consists of two components: a family-based component (FC) and a preschool-based component (PC). The present study will primarily focus on the FC. A detailed description of other elements of the SuperFIT intervention approach and research protocol can be found elsewhere [28].

The SuperFIT FC aimed to integrate healthy nutrition and physical activity into children’s daily lives within the home setting. It was complementary to the PC of SuperFIT. Children and parents attended sessions after preschool hours. Other caregivers such as aunts or grandparents were also able to attend FC sessions, although this only occurred sporadically. Direct parental involvement was incorporated in the form of parent, child, and family sessions. Parent sessions aimed to increase knowledge about nutrition, PA, and positive parenting, based on Lifestyle Triple P seminars [39], which included three interactive lectures. Group interaction was stimulated, enabling parents to share their experiences. Special attention was given to the influence of the different types of environment and strategies to change these (e.g., setting the right example, availability of fruit at home). At the same time, children attended separate child sessions in which they played activity games or prepared healthy treats. Finally, parents and children attended family sessions together. Family sessions entailed activities to experience healthy nutrition and PA together as a family, with the primary focus on having a positive experience related to EBRBs. A second aim was to inspire parents by showing them simple activities using several simple materials to promote PA and stimulate motor skills, or experiencing and tasting new types of fruits and vegetables. All sessions were given by trained implementers (*n* = 5) with expertise in nutrition, PA, behavior change, or certified Triple P trainers. In addition, indirect parental involvement was stimulated in the form of factsheets and newsletters. Factsheets provided summaries of the information discussed during parent sessions. Newsletters informed parents of SuperFIT intervention activities organized at preschools, as part of the PC.

### 2.3. Timeline

The implementation of SuperFIT took place between April 2017 and May 2018 (Figure 1). Three rounds of the FC were organized. The first round (starting in May 2017) consisted of seven sessions—four family sessions and three simultaneous parent and child sessions. However, as seven sessions were considered to be too demanding according to parents and implementers, the second (starting in September 2017) and third (starting in January 2018) round consisted of five sessions—two family sessions and three simultaneous parent and child sessions. Baseline measurements (T0) were performed before the start of the intervention activities from January to April 2017. Two follow-up measurements were performed in November/December 2017 (T1) and May/June 2018 (T2). In-depth interviews were collected continuously during the implementation period.

### 2.4. Study Population

The study population differed across research activities. All children (*N* = 427) attending the twelve intervention preschools were exposed to the PC. All parents were invited to participate in the overall study of SuperFIT. At least one parent of the child had to be able to understand Dutch, and both parents had to sign the informed consent. Parents of 93 children (22%) agreed to participate. In addition, parents were invited to participate in the complementary FC and additional research. Forty-one of the 93 parents agreed to participate (44%). A second recruitment round included six additional parents for the FC. All 47 parents participating in the FC received questionnaires and 15 parents agreed to participate in an additional interview. Two implementers of the FC and one implementer of SuperFIT in general participated in interviews (*n* = 3) as part of a general process evaluation of SuperFIT and their views on the FC.

### 2.5. Measurements

Methods and results were structured using the RE-AIM framework (see Table 1). No long-term assessments were performed within the theme of maintenance, and effectiveness is described elsewhere [29,30].

#### 2.5.1. In-Depth Interviews

Semi-structured interviews were conducted by two researchers (CL and TA), with 15 parents during or directly following their participation in the FC. Interviews addressed recruitment (reach), reasons to participate (adoption) and appreciation, strengths and limitations of the FC (implementation). Parents received questions about perceived changes in their own behavior, the home setting, or their child’s behavior after participating in the FC (impact). Two researchers conducted the interviews.

In addition, in-depth semi-structured interviews were conducted by one researcher (CL) with three trained implementers with in-depth insight of the FC and SuperFIT in general. Questions concerned the recruitment (reach) and their experiences implementing the FC (implementation). Implementer interviews were conducted between the first and second rounds of the FC implementation.

All interviews were in Dutch and were audio-recorded. Participants provided verbal consent at the start of the interview. Topic lists for each interview round are included in Appendix B, Appendix C and Appendix D.

#### 2.5.2. Observations of FC Intervention Activities

Parent and family sessions were observed using a free-form protocol. Events of note that occurred during implementation were recorded as well as and the general impression of the session (contextual factors). The same researcher (CL) conducted observations for all rounds and sessions. Family attendance of all sessions was also recorded. Attendance of all sessions combined was considered to represent the ability of the FC to reach families. In addition, variety in attendance over time was seen as adoption of the FC.

#### 2.5.3. Questionnaires

All 93 parents participating in the larger research study regarding SuperFIT in general were asked to fill in a questionnaire including parent demographics and background variables (age, educational level, employment status, and country of birth). Furthermore, the 47 parents participating in the FC received additional questionnaires. These assessed nutritional and PA-related parenting practices (impact) based on the Comprehensive Feeding Practices Questionnaire (CFPQ) [41] and the Preschooler Physical Activity Parenting Practices questionnaire respectively (PPAPP) [40]. Nutritional scales included ‘Encourage balance and variety’, ‘Environment’, ‘Involvement’, ‘Modeling’, ‘Monitoring’, ‘Teaching about nutrition’, ‘Emotion regulation’, ‘Food as reward’, and ‘Child control’. PA scales included ‘Parental engagement’, ‘Promote screen time’, ‘Promote inactivity’, ‘Psychological control’, and ‘PA restriction’. Example questions include ‘*Do you encourage your child to eat healthy foods before unhealthy ones?’* and ‘*How*
*often do you go on a walk with your child?*’. Parenting practices were assessed on a 5-point Likert scale, ranging from never (1) to always (5), or totally disagree (1) to totally agree (5). Both questionnaires were adapted to the Dutch setting through changes in examples or by eliminating questions that were not applicable. In addition, parents were asked to fill in a parental self-report of the physical home environment, based on The Environment and Policy Assessment and Observation-Self Report (EPAO_SR) [42]. The EPAO_SR was originally developed for childcare centers and adapted to the Dutch home setting. It assesses the indoor and outdoor environment such as the layout of a garden, and the availability of PA equipment such as balls or hoops. Parents were first asked to score availability as ‘yes’ and ‘no’, and then to tick the boxes indicating whether this equipment was ‘available inside the house’ and ‘available in the garden’. These over-time measurements of parenting practices and the physical home environment represent the (perceived) impact of participating in the FC. At the follow-up measurements, parents were also asked to score their appreciation of the SuperFIT FC on three themes: content of the intervention, design of the implementation, and impact of participation. Example questions were whether sessions were informative, how they evaluated the group size, and whether the intervention helped them to change behavior.

### 2.6. Data Processing and Analysis

Qualitative data were transcribed verbatim, anonymized and then imported in NVivo 12.0 software (QSR International, Doncaster, Victoria, Australia). An inductive research approach was used as the coding tree was not solely theory-driven but emphasized the chronological structure of the data. Themes were roughly defined from the beginning: reach, adoption, implementation, and perceived impact of SuperFIT. Nodes such as ‘before participation’ or ‘after participation’ and ‘parent’, ‘home setting’ or ‘child’ impact provided structure. Open coding was used, enabling an open view toward the data. Two researchers (LH and FM) independently analyzed 20% (three of 15) of the interviews through interim consensus meetings to reach a final coding tree and several decision rules. One of the two researchers then analyzed the remaining interviews (LH 47% and FM 33%) using the final coding tree. A final meeting was held to discuss and resolve any uncertainties during final analysis.

Results are described according to their chronological order in the implementation process, in which key observations from both qualitative and quantitative data will be highlighted. Themes were based on the coding tree and RE-AIM framework [37] (Table 1). Reach (whether SuperFIT was able to reach parents), adoption (how SuperFIT was delivered and adopted), and implementation (experiences with the implementation process from the perspective of parents and implementers). In addition, parents were asked whether SuperFIT provoked changes in their daily life routines. This was categorized as ‘perceived impact’ on three different levels: parent, physical home environment, and child EBRBs.

All quantitative data from questionnaires were entered and cleaned in SPSS Version 25.0 (IBM, Armonk, NY, USA). Items were recoded if necessary; the highest score always reflected the most positive value for that question to accommodate the study population (i.e., low literate participants). Parenting practices scales were calculated as the sum of the items divided by number of items included in the scale. Cronbach’s alpha was used to test scale reliability of the scales at T0. A Cronbach’s alpha greater than 0.50 was considered acceptable [43]. Items were deleted from the scale whenever this improved the Cronbach’s alpha to greater than 0.50. Deleted items and the items of unreliable scales were analyzed as single items. Descriptive analyses were performed for each measurement point (i.e., T0, T1, T2). Missing data and the small sample sizes (*n* = 47) limited the ability to perform further statistical testing.

## 3. Results

### 3.1. Reach

#### 3.1.1. Participants of the FC

The FC of SuperFIT reached 47 families, which is about half of the families participating in the research activities of SuperFIT in general. However, this represents only 11% of all children who had been exposed to SuperFIT at an intervention preschool. The majority of participating parents were female, employed, and born in the Netherlands (see Table 2 for characteristics of the included participants). Parental mean age in years was around 33–34 years in the different samples. Implementers were on average 30.3 years of age and worked as either a lifestyle coach or PA and health coach. They implemented the family and child sessions of the FC (*n* = 2) and were all involved with the development and implementation of SuperFIT in general.

#### 3.1.2. Communication

The FC of SuperFIT was promoted to parents as ‘*a fun way to discuss the topics*’, ‘*playfully dealing with nutrition*’, and ‘*exercising together with your child*’. To promote SuperFIT, the childcare organization and research team distributed information flyers, preschool locations displayed posters, and a kick-off event was organized for parents. In addition, researchers and implementers visited preschool locations to explain SuperFIT verbally. However, the majority of interviewees could not recall specifically how they had been notified of SuperFIT. Frequently mentioned responses were ‘childcare informed us’ or ‘we received a letter or newsletter’. Parents indicated that the information flyers contained a large amount of information including details on participating in medical scientific research. Parents thought that it might have deterred people from participating. Implementers expressed the same concern: recruitment materials could have been a better fit with the target group, “*It was difficult to read through the scientific terms and quite extensive information*” (implementer).

#### 3.1.3. Attendance

Based on the attendance forms, a session reached on average 60% of the families. In general, family sessions had better attendance rates (average = 66%, range 45–80%) compared to parent sessions (average = 55%, range 22–88%).

### 3.2. Adoption

Parents received tools and practical tips to apply at home. Adopters expressed having a variety of expectations prior to their participation. This ranged from receiving tips and tricks concerning nutrition to SuperFIT being a fun physical activity event for their child to participate in such as, “*How to involve my daughter in meal preparation, to receive some practical tips on that*” (parent). Parents had not expected the parent session on positive parenting. According to the respondents, clarifying the content and design of the sessions would perhaps have increased interest in participating among fellow parents who did not participate. The large time investments on the part of parents increased interest in the study results and desire for personal feedback (i.e., how am I doing compared to other parents?). Attendance rates varied over the different FC rounds and over sessions (range 22–88%). During the first round, attendance rates decreased as the sessions progressed.


*Non-adopters of the FC chose ‘no time’ and ‘no need’ most frequently in the questionnaire as reasons for not participating. A small percentage (13%) also indicated that the planning of the FC sessions conflicted with their child’s nap, one of the multiple-choice options. In interviews, both parents and implementers suspected that families who would have benefitted the most from participation (e.g., low SES, unhealthy lifestyles) were not participating.*


### 3.3. Implementation

#### 3.3.1. Content

Parents appreciated the nutrition and PA-related content and felt that this appealed more to parents than the session on parenting. Quantitative scores indicated the FC to be fairly interesting (mean = 3.84, range 2–5), fairly educational (mean = 3.84, range 3–5), and clear (mean = 4.32, range 4–5). Activities concerning healthy nutrition or physical activity were all considered good (mean = 4.16, and mean = 4.42, range 3–5 for both).

It emerged that the information provided on nutrition and PA was not new to parents. Nevertheless, they appreciated the repetition and emphasized that all information was applicable to young children. “*I think there is too much attention on it in general, we’ve heard it a hundred times already*” (parent). This was in contrast to several observations. “*Parents appear to hardly have the knowledge to be able to read food labels and are not really familiar with the dietary guidelines*” (observer). Observations described the surprise of parents regarding, for instance, the daily recommended amount of vegetables for young children or knowing what a product contained by reading food labels. Observations also indicated that parents were interested in learning about food labels. However, parents did not highlight this activity during the interviews.

#### 3.3.2. Design of the FC

In interviews, parents indicated that they greatly appreciated the family sessions for various reasons. First, it allowed them to receive on-the-spot guidance or coaching on their role as a parent during PA activities. “*What I really appreciated was the exercise activity together with my daughter. Especially to see what you pointed out on how to stimulate PA behavior of your child and how to guide them*” (parent). Observations confirmed that the PA sessions especially induced great parental participation, enabling the PA implementer to provide tailored feedback. Nutrition sessions evoked a more passive participation, in which parents mainly listened. Second, their child’s enthusiasm was contagious. “*He ate fruit that he never ate at home before*” (parent). Implementers endorsed this in their interviews, “*That [enthusiasm] also has an impact on the parents… if a child asks ‘when is the next session?’, that influences s a parent’s attitude*” (implementer). This was reflected in the scoring of group session appreciation from their child’s perspective (mean = 4.63, range 3–5). Finally, parents indicated that the family sessions added to the variation of the program.

A limitation that was mentioned was the lack of group interaction during the parent sessions. “*What I regretted was the little input that the group of parents had during the group sessions*” (parent). They specifically expressed their need to hear practical knowledge and personal experiences from peers. It is unlikely that this was attributed to the implementer, since all implementers were positively rated both quantitatively (mean range = 4.26–4.68, range 3–5) and qualitatively. Observations also described the implementers’ struggle to get parents to respond. Implementers tried different interactive questions such as ‘Can you share some of the challenges you face in parenting?’ or ‘Do you recognize this?’ and used statements such as ‘I have clear rules for screen time’ to which parents had to respond. Small assignments such as ‘Write down your child’s activity pattern’ or discussing food labels did seem to promote parental interaction, according to observations. One implementer stated that parents were given the opportunity to interact during parent sessions. “*At the parent session they are able to ask questions or exchange experiences*” (implementer). Another implementer noted a difference between groups, but was unable to pinpoint underlying causes. “*Parents in the second group were more involved, seemed more interested, had a more active attitude*” “*No idea [why], could be anything*” (implementer).

The number of FC sessions was reduced from seven to five after the first round of sessions, and the day and time were changed in order to enhance participation levels. Parents recognized the scheduling issues, as each family has to attend a variety of activities (i.e., sports, visiting grandparents). Still, planning and frequency of the sessions were not considered as a major barrier for the participating respondents. They considered their participation in research activities to be a larger time investment than participation in the intervention.

### 3.4. Impact

#### 3.4.1. Parents

Parents often reflected on themselves when we asked them about the impact of participating in the FC. “*The awareness part, like yes we have to pay attention to that and we have to work on that*” (parent). Several parents labelled the family sessions as eye-opening. Seeing their child try new sorts of fruit and vegetables, or playing easy PA games made them realize that change might be relatively easy to achieve. “*I’ve learned that… it is actually very easy to be physically active inside the house too*” (parent). However, others labelled their participation as being merely a confirmation of their existing healthy behaviors.

Parents considered the guidance they received during parent–child activities to be a strong point of the FC as it further increased their understanding on how to stimulate healthy behaviors. A number of respondents recognized an increased awareness and usage of positive parenting, often in the context of dinner and offering new fruits and vegetables. “*If he says ‘I don’t like it’ we still try to stimulate him but we don’t get angry, thinking we’ll try again next week*” (parent). This positive change was also reflected in the quantitative data on parenting practices of the FC participants (Table 3). Notable favorable changes after the intervention were seen for providing a healthy food environment, monitoring of children’s dietary intake, and using food as a reward. Restriction of physical activity due to safety concerns increased over time. Data seemed to indicate high scores for several parenting practices at baseline already. Concerning the nutrition scales, the majority of parents were at the higher end of the scale (4–5) for ‘encourage balance and variety’ (75%) and ‘modeling’ (85.7%). The converse was also true: 75.1% of the parents were already at the lower end of the scale (1–2) for emotion regulation at baseline. This is in line with parents generally indicating that they were already doing fine before participation.

#### 3.4.2. Physical Home Environment

In contrast to the notable changes in parenting practices, parents were not able to pinpoint major changes in the physical home environment (Appendix A). Family session activities aimed to show that physical activity could be promoted using relatively simple materials (e.g., bean bags, ball). Concerning nutrition, parents were informed about the importance of making a variety of fruit and vegetables available. Parents, however, indicated no urgency in integrating these strategies at home. A frequently mentioned reply was the existing high availability of fruit at home before the intervention, although parents did try to present fruit differently. “*Fruit was never really a problem, but we do prepare it more often together. Smoothies for instance… we make things like that even more now*” (parent).

Most parents indicated that they did not buy new PA equipment or new kinds of fruit. “*I’m thinking about buying one of those hula hoops and bean bags for her… haven’t done that yet*” (parent). The amount of PA equipment at home did not change notably (Appendix A).

#### 3.4.3. Child EBRBs

Almost all respondents indicated observing no alterations in their child’s eating or physical activity behavior, in both qualitative and quantitative data. They did, however, recognize the increased variety in fruits and vegetables their child consumed at childcare. Parents themselves stated that their child already had a healthy activity or eating pattern before the start of the intervention. “We were satisfied with that [child’s diet] already” (parent).

However, asking parents directly to reflect on the content of SuperFIT led them to express observing effects in their child’s behavior. This was often intertwined with stories about their child’s enthusiasm for SuperFIT. “*She says: ‘I have to eat vegetables, then I’ll get bigger’*” (parent). Children liked the activities such as making a fruit animal or stretching exercises, a part of the family and child sessions. Children’s enthusiasm led them to perform the activities themselves or to ask their parents whether they could make the fruit animals again “*She is doing the exercises that she evidently performed with you [SuperFIT]*” (parent).

## 4. Discussion

The present study provided a process and impact evaluation of the FC within the home setting. For this purpose, we applied the RE-AIM framework [37]. We describe the results per concept of this framework.

The FC of SuperFIT was able to reach roughly half of the total number of research participants and only one tenth of all children attending an intervention preschool. The reach of the FC is therefore considered to be low. Moreover, it is arguable whether the intended vulnerable target population was sufficiently reached. Although participating intervention preschools were selected based on the low socio-economic status of the preschool neighborhood, relatively many highly educated parents participated in the FC. This is in line with previous research pointing out the challenge of reaching and engaging the families who need support the most [19,34,44]. Reaching and engaging, specifically those families, is of utmost importance in efforts to reduce health inequalities [45]. A lot is still to be learned about how to optimally reach and involve parents [23]. A hindering factor may have been SuperFIT’s recruitment materials, which appeared difficult to read. Unfortunately, some of these materials are mandatory information from medical ethical committees, and can therefore not be modified. Still, future recruitment strategies should balance the required information provision with participant needs, for instance, by using tailored communication and pictures [19] or by engaging parents in research development [46].

Concerning adoption, participating parents expressed different motives for their decision to participate. Most notable was the need to receive practical tips, specifically on how to stimulate their child’s healthy EBRBs. Non-adopters identified lack of time or need as a reason for not participating in the FC. According to parents, clarifying the content and design, specifically mentioning the fun activities with their child, would have increased interest to participate in the FC. Similarly, other researchers found the relevance of an intervention to be of importance for engagement [35]. Parents participating in the intervention for a specific need, or because they perceived the intervention as having a focus on their family’s everyday life, further facilitates active engagement [35].

Overall, parents and children evaluated the content of the FC positively. Receiving on-the-spot guidance while seeing their child’s enthusiasm during family activities was greatly valued. This is in line with, for instance, the Healthy Dads Healthy Kids intervention, in which children’s enthusiasm for father–child activities is hypothesized to reinforce a shift in family lifestyle [47]. Having fun and receiving tailored feedback therefore may be crucial for the success of a family intervention. The general information concerning nutrition and PA was perceived as not being new, even though sessions actively also included practices to be performed at home, addressing skills and self-efficacy. This corresponds to studies stating that although many preschool parents understand the importance of EBRBs, they encounter barriers to routinizing these practices [48,49].

Regarding perceived impact, parents indicated having an increased awareness and understanding of their own behavior as well as their role in their child’s behavior. Quantitative data on parenting practices confirmed this. Favorable changes over time were seen for various practices. This is interesting, considering the fact that parents themselves were not able to see the benefit of parenting sessions, preferring information on nutrition and PA. Promoting positive parenting therefore remains an effective and pivotal part of family interventions [39] for the small changes it unconsciously initiates. In addition, for some practices, greater changes were seen at the first follow-up compared to the second follow-up. Implementation of the FC or related activities occurring at intervention preschools might have caused a short-term increase in awareness of their own behavior. This suggests that maintenance of new behaviors might play a role. Parents might need sustainable incentives or recurring intervention activities to prolong the short-term positive change.

Parents did not express an urge to change the physical environment and did not perceive any changes in their child’s behavior. This suggests that the tipping point at which parents are actively translating learned strategies into implementation at home had not yet been achieved with SuperFIT. Several factors potentially play a role in this gap between knowledge and behavior. Research states that to improve EBRBs in families, intervention strategies should include easy-to-implement behavior change strategies [50]. However, family sessions within the FC specially focused on the provision of those strategies. The FC might need to address ‘know how’ knowledge further [51]. Other researchers emphasized sharing experiences and challenges with peers (e.g., other parents) [52], as this would aid translating knowledge into behavior at home [49]. Parents in the present study confirmed this by expressing a need to hear personal experiences from peers. Despite efforts to increase group interaction, parents felt that this happened insufficiently in the parent sessions. How to effectively stimulate interaction among parents is something that will need to be further addressed for the further development of the FC.

Finally, several parents indicated that their families already had a healthy lifestyle. This might explain a lack of urgency or perceived need for parents to actively implement new strategies. Self-reported quantitative findings seem to confirm that most families were already using health-promoting practices, although only for some outcomes. In contrast, effect evaluations of SuperFIT using objectively measured child outcomes did not show high baseline scores [29,30]. Nonetheless, children were enthusiastic about the activities learned and also performed them at home. Similar take-home effects have been seen in older children before and have the potential to lead to long-term changes [53]. Future research should determine parental needs to optimize parental involvement in obesity prevention interventions.

To sum up, interventions should use easy-to-read or graphic recruitment materials, clearly indicate what the program entails, focus on fun family activities, provide on-the-spot guidance, and facilitate peer-to-peer discussion. This requires a time and energy investment on the part of families as well as an interest and a perceived need to change. SuperFIT made use of direct parental involvement in the FC, which is the assumed favorable form of involvement [21,22,23]. However, this more intensive form of parental involvement might result in compromises regarding reach and adoption, as fewer parents might participate. Furthermore, it remains unclear whether direct strategies are more effective in creating intervention enactment among parents [18]. Simply implementing any form of direct (or active) parental involvement is therefore not the solution. Parental involvement should reflect an optimal balance between intensity, accessibility, and content for parents. More research is needed to establish such an optimal balance.

### Strengths and Limitations

The present study adds to our knowledge on parental involvement using a mixed methods approach including the perspective of parents and implementers. Combining quantitative (questionnaires among participating parents in the FC) and qualitative (interviews with parents and implementers, and observations) research allows for data triangulation and the reflection through different perspectives. This provides a better understanding of the implementation of the intervention and subsequent changes in the home setting. Using concepts such as reach, adoption, and implementation enabled a structured exploration of the potential impact of SuperFIT [37]. Although we made use of RE-AIM, effectiveness and maintenance were outside the scope of the present study. In addition, as SuperFIT is an intervention approach tailored to the context, we modified the assessment of implementation, as concepts such as fidelity and completeness do not apply. We therefore aimed to assess implementation in terms of qualitative appreciation.

There were also some limitations to this study. Due to choices made to limit participant burden, no quantitative data were available from the control group. We were therefore not certain that the observed changes in the home environment were solely attributable to SuperFIT. Societal changes such as a general increased interest in a healthy lifestyle may have contributed to the changes mentioned in this study. In contrast, parents might have been unaware of the initiated changes and consequently failed to report them. In addition, missing data and relatively small sample sizes in the quantitative measures limited the ability to perform statistical testing. Finally, the use of specific questionnaires may need to be reconsidered. Although standardized and validated questionnaires were used to measure perceived impact on the physical environment [42] and parenting practices [40,41], it is debatable whether these questionnaires were sensitive enough to detect changes. For example, questions only concerned the presence of play equipment available indoor or outdoor (‘yes’ or ‘no’). Changes made in equipment sorts, increased availability within an equipment category, and whether parents actively used the equipment were not included. This should be taken into account when interpreting the findings.

## 5. Conclusions

A well-designed intervention can help parents to promote EBRBs in their preschool-aged child. We recommend that health professionals balance content with accessibility and intensity for their participants. This aids in the reach, adoption, implementation, and potential impact on child EBRBs.

## Figures and Tables

**Figure 1 nutrients-13-01605-f001:**
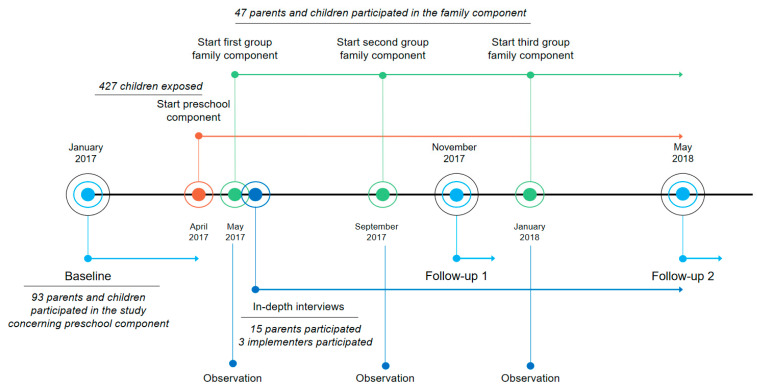
Visual representation of the timeline of all research and intervention activities concerning the SuperFIT approach.

**Table 1 nutrients-13-01605-t001:** Thematic structure of the present paper.

Themes		Data Source		Concept	Example Question(s) or Content
1	Reach	Qualitative	Parental interviews	Communication Recruitment Strengths and limitations	*How did you come into contact with SuperFIT?* *How were you informed about the FC?*
			Implementer interviews	Recruitment Strengths and limitations	*What were your thoughts on the promotion of SuperFIT? (kick-off session, information folder)*
		Quantitative	Registration forms	Attendance	Number of parents attending the FC intervention activities.
			Parental questionnaires	Demographics	Characteristics of participants of the FC.
2	Adoption by parents	Qualitative	Parental interviews	Expectations Reasons to participate	*What made you decide to participate?*
	Non adoption by parents	Quantitative	Questionnaires for PC participants	Reasons to decline participation	*What was your reason for not participating in the FC? (multiple choice)*
3	Implementation				
3a	FC content	Qualitative	Parental interviews	Appreciation of FC intervention activities Strengths and limitations	*Can you give an example of something that you liked or disliked during the sessions? Which part or parts appealed to you the most, and why? What did you think of the materials used?*
			Implementers interviews	Experiences	*How did you experience the first family session?*
			Observations	Contextual factors	*Group dynamics, questions asked*
		Quantitative	Parental questionnaires	Appreciation of FC activities (scoring)	*Did you find the program educational?* *What did you think of the activities concerning nutrition?*
3b	FC design	Qualitative	Parental interviews	Appreciation of the FC set-up, Strengths and limitations	*What did you think of the intervention design, the various session types? What did you think of the timing and number of sessions?*
			Implementers interviews	Experiences Strengths and limitations	*How would you describe the cooperation with the parents?*
			Observations	Contextual factors	
		Quantitative	Parental questionnaires	Appreciation of the FC set-up (scoring) Duration Time investment	*What did you think of the duration of the program?*
4	Impact				
4a	Parent(s)	Qualitative	Parental interviews	Self-reported impact on own behavior	*Do you feel that there has been a change in the way you deal with your child’s (healthy) EBRBs?*
		Quantitative	Parental questionnaires	PA parenting practices (PPAPP) Nutritional parenting practices (CFPQ)	*How often do you go on a walk with your child?* *Do you encourage this child to eat healthy foods before unhealthy ones?*
4b	Physical home environment	Qualitative	Parental interviews	Self-reported impact on family daily life or home setting	*Were you able to use the received information at home? How? Why not?*
		Quantitative	Parental questionnaires (EPAO_SR)	Availability of play equipment	*Which of the following play equipment is available inside the house?*
4c	Child EBRBs	Qualitative	Parental interviews	Self-reported impact on child behavior	*Do you feel that there are changes in the behavior of your child due to SuperFIT? What was the situation before participating in SuperFIT?*

FC = Family-based Component, PC= Preschool-based Component, PPAPP = Preschooler Physical Activity Parenting Practices questionnaire respectively [40], CFPQ = Comprehensive Feeding Practices Questionnaire [41], EPAO_SR = The Environment and Policy Assessment and Observation-Self Report [42].

**Table 2 nutrients-13-01605-t002:** Participant samples and demographics.

Demographics	Questionnaires on Parenting Practices	Questionnaire on Quantitative Evaluation of FC	Interviews
	*n* (%) ^1^	*n* (%) ^1^	*n* (%) ^1^
Parents *n* (%) ^1^	41 (100)	19 (100)	15 (100)
Gender, female		16 (84.2)	13 (86.7)
*T0*	25 (80.6)		
*T1*	20 (83.3)		
*T2*	23 (92.0)		
Number of children			
*-one*	14 (45.2)	4 (26.7)	4 (26.7)
*-two*	15 (48.4)	9 (60.0)	8 (53.3)
*-three*	2 (6.5)	2 (13.3)	3 (20.0)
Educational level ^2^			
*-low*	5 (12.5)	1 (5.3)	1 (7.1)
*-middle*	12 (30.0)	8 (42.1)	5 (35.7)
*-high*	23 (57.5)	10 (52.6)	8 (57.1)
Employment status			
*-unemployed*	10 (32.3)	3 (20.0)	3 (20.0)
*-employed (full-time or part-time)*	21 (67.7)	12 (80.0)	12 (80.0)
Country of birth			
*The Netherlands*	37 (92.5)	18 (94.7)	n.a.
Implementers *n* (%)			3 (100)
Percentage of females			2 (66.7)
Educational level			
*-high*			3 (100.0)
Employment status			
*-part-time*			2 (66.7)
*-full-time*			1 (33.3)
Country of birth			
*-Netherlands*			3 (100.0)

FC = Family-based component, n.a. = not assessed. ^1^ Due to missing data, *n* can vary; percentages are based on available data. ^2^ Based on ISCED classification.

**Table 3 nutrients-13-01605-t003:** Nutritional and PA-related parenting practices of participants in the FC, group differences.

Scale	Items	Cronbach’s Alpha (T0)	Baseline (*n* = 28)	Follow-Up 1 7–10 Months (*n* = 22)	Follow-Up 2 13–16 Months (*n* = 25)
			Mean (SD)	Change in Mean	Change in Mean
**Nutritional parenting practices ^1^**					
***Scales that promote healthy EBRBs***					
Encouraging balance/variety	4	0.82	4.38 (0.68)	+0.10	+0.03
Providing a healthy food environment	3	0.56	3.30 (0.81)	+0.35	+0.18
Involving child with food	3	0.63	4.05 (0.77)	+0.01	+0.14
Modelling healthy food intake	4	0.84	4.38 (0.81)	+0.03	+0.14
Monitoring diet child	2	0.74	3.48 (1.12)	+0.38	+0.24
Teaching child about nutrition	4	0.66	3.73 (0.89)	+0.36	+0.28
***Scales that inhibit healthy EBRBs healthy EBRBs***					
Using food for emotional regulation	3	0.72	1.73 (0.65)	−0.18	−0.11
Using food as reward	3	0.61	2.69 (0.92)	−0.43	−0.54
Pressuring to eat	3	0.60	3.22 (0.83)	+0.26	−0.05
***Single items that promote/inhibit healthy EBRBs***					
Child allowed to choose between different healthy products (*environment*)			3.75 (1.04)	+0.34	+0.13
Child pressured to finish plate (*pressure*)			2.14 (1.21)	−0.23	+0.18
Child allowed to choose food (*control*)			3.04 (0.79)	+0.01	+0.20
Child allowed to choose food served (*control*)			3.29 (0.81)	+0.12	+0.59
Preparing other food for child (*control*)			2.25 (1.14)	+0.07	−0.33
Child allowed to snack (*control*)			2.71 (0.90)	−0.53	−0.11
Child allowed to leave table early (*control*)			2.27 (1.16)	+0.23	+0.41
**PA parenting practices ^1^**					
***Scales that promote healthy EBRBs***					
Parental engagement	13	0.87	3.61 (0.51)	+0.12	+0.10
***Scales that inhibit healthy EBRBs***					
Promoting screen time	2	0.61	2.75 (0.71)	−0.05	−0.25
Using psychological control of PA	5	0.60	2.10 (0.52)	−0.03	−0.06
Restricting PA	3	0.72	1.68 (0.73)	+0.19	+0.33
***Single items that promote/inhibit healthy EBRBs***					
Child allowed to play outside (*restriction*)			3.93 (0.66)	+0.07	+0.27
Availability of play equipment outside (*engagement*)			4.57 (0.63)	−0.25	+0.11
Carrying child (*promote inactivity*)			2.41 (0.81)	−0.23	−0.57
Using stroller (*promote inactivity*)			2.14 (1.08)	−0.23	−0.54
Using the car while distance is walkable (*promote inactivity*)			2.71 (0.90)	−0.30	−0.35

EBRBs = Energy Balance Related Behaviors. SD = standard deviation, PA = physical activity. ^1^ Scored on a five-point scale ranging from 1 = Never or Totally Disagree to 5 = Always or Totally Agree.

## Data Availability

Datasets used and/or analyzed during the current study are available from the corresponding author on reasonable request.

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
