# Peer review of "Involving Parents in Promoting Healthy Energy Balance-Related Behaviors in Preschoolers: A Mixed Methods Impact and Process Evaluation of SuperFIT"

_nutrients, 2021, doi:10.3390/nu13051605_

Round 1
Reviewer 1 Report
This is a well written paper on an important topic, using mixed methods to evaluate the SuperFIT intervention. This seems like a well designed and well thought out intervention, using systems theory to target important settings, involving both parents and childcare workers.
- The introduction is well written - the authors refer to other interventions and reviews conducted, however they should state what were the results of other interventions, ie. were they mostly successful or not? What outcomes did they measure? In the discussion they should then compare their findings with othe similar interventions.
- The use of the REAIM model is appropriate for evaluation of the intervention. However -Can the authors please explain this sentence and how this constitutes reach? “Family attendance of all sessions was also recorded (reach.)” This seems more like fidelity/adherence to the program?
- “the highest score always reflected the most positive value for that question” – there is a danger of participants answering all with the same value when all items follow the same direction (thus reverse scoring is sometimes used)
- Results – in both table 3 and table S1 I see that for some measures there was a greater change at the first follow up compared to the second follow up – could the authors expand on this/suggest some interpretation? (Possibly to do with maintenance of new behaviours?) Also, could they add significance for the change?
- The authors find, as is common and a problem of many health interventions, that it is difficult to reach the most in need population – how were the intervention preschools chosen? Could this have affected the demographic of participants? Perhaps the same program in a lower SES population would have had a greater effect?
The qualitative analysis adds useful information to the program evaluation, especially of use to those planning future interventions.
Author Response
We want to thank the reviewer for the compliments and constructive suggestions to further improve the present manuscript.
Please see the attachment for our Review Report.

Reviewer 2 Report
Thank you for the opportunity to review the paper presented by the authors. The authors present the results of the project evaluation and the use of mixed methodology in the evaluation of the effectiveness of an intervention programme for parents of young children. The intervention presented is grounded in ecological systems theory emphasizing the role of multiple proximal and distal developmental environments and their interactions on child functioning. I believe that the paper is part of a trend worth promoting of analysing the results of intervention programmes, taking into account a comprehensive assessment of various elements that may modify their effectiveness. I agree with the authors' conclusion that effective interventions should represent a balance between different areas, such as programme content, intensity of activities and accessibility. The analysis confirms that when planning a prevention programme, attention should be paid not only to areas directly related to the focus of the intervention, but also, for example, to factors that potentially increase satisfaction with participation in the programme. Indicators which are related to the process of programme implementation and the results obtained during its evaluation can represent changes in motivational and environmental factors which indirectly influence health behaviour.
Summarizing, the paper is prepared very well. All parts of the manuscript are described in a fair and very comprehensive way. In my opinion the paper should be accepted for publication in its current form.

Author Response
We thank the reviewer for the compliments and kind words.
Please see the attachment for our complete Review Report, including our responses to Reviewer 1.
